# A Novel Tanshinone Analog Exerts Anti-Cancer Effects in Prostate Cancer by Inducing Cell Apoptosis, Arresting Cell Cycle at G2 Phase and Blocking Metastatic Ability

**DOI:** 10.3390/ijms20184459

**Published:** 2019-09-10

**Authors:** Mengling Wang, Xueyi Zeng, Shengyou Li, Zekun Sun, Jia Yu, Chao Chen, Xiangchun Shen, Weidong Pan, Heng Luo

**Affiliations:** 1State Key Laboratory of Functions and Applications of Medicinal Plants, Guizhou Medical University, Guiyang 550014, China; 2College of pharmacy, Guizhou Medical University, Guiyang 550025, China; 3Key Laboratory of Chemistry for Natural Products of Guizhou Province and Chinese Academy of Sciences, Guiyang 550014, China; 4College of pharmacy, Guizhou University, Guiyang 550025, China

**Keywords:** quinone analog, prostate cancer, cell apoptosis, cell cycle arrest, metastasis potency

## Abstract

Prostate cancer (PCa), an epithelial malignant tumor, is the second common cause of cancer death among males in western countries. Thus, the development of new strategies is urgently needed. Tanshinones isolated from *Salvia miltiorrhiza* and its synthetic analogs show various biological activities including anticancer effects. Among them, the tanshinone analog 2-((Glycine methyl ester)methyl)-naphtho (**TC7**) is the most effective, with better selectivity and lower toxicity. Therefore, in this work, the effect of **TC7** against PCa was investigated through assessing the molecular mechanisms regulating the growth, metastasis, and invasion of PCa cells. Human PCa cells, PC3 and LNCAP, were used to evaluate **TC7** mechanisms of action in vitro, while male BALB/c nude mice were used for in vivo experiments by subjecting each mouse to a subcutaneous injection of PC3 cells into the right flank to evaluate **TC7** effects on tumor volume. Our in vitro results showed that **TC7** inhibited cell proliferation by arresting the cell cycle at G2/M through the regulation of cyclin b1, p53, GADD45A, PLK1, and CDC2/cyclin b1. In addition, **TC7** induced cell apoptosis by regulating apoptosis-associated genes such as p53, ERK1, BAX, p38, BCL-2, caspase-8, cleaved-caspase-8, PARP1, and the phosphorylation level of ERK1 and p38. Furthermore, it decreased DNA synthesis and inhibited the migration and invasion ability by regulating VEGF-1 and MMP-9 protein expression. Our in vivo evidence supports the conclusion that **TC7** could be considered as a potential promising chemotherapeutic candidate in the treatment of PCa.

## 1. Introduction

With the rapid development of medical technology, a variety of cancers have been rapidly and accurately diagnosed and treated in the early stage, thus significantly extending the life of most patients [1]. However, recurrence and drug resistance in cancer patients after surgery is a serious challenge in cancer treatment. Hormone-dependent cancers including prostate cancer (PCa) and breast cancer, involve the secretion and regulation of various hormones, leading to their imbalance and causing many fatal complications that seriously affect the treatment of such patients, with a high death rate [2]. Since these hormone-dependent cancers are highly metastatic, early diagnosis and treatment are more difficult, which means that the control of metastasis in these cancers represents a key aspect in the improvement of the survival rate of cancer patients after surgery [3,4,5,6]. Thus, the control of the rapid proliferation, migration, and invasive ability of these highly metastatic cancer types has become a major scientific issue and research hotspot. 

PCa, as a malignant tumor with a high incidence in males, remains an important public health concern and is the second leading cause of cancer-associated mortality in Western countries due to a higher incidence of metastasis [7], with more than 1.2 million novel cases diagnosed every year worldwide [8,9]. Age, race, family history, genetics, and obesity are identified as the risk factors for PCa [10]. At present, early-stage PCa can be treated via surgery or radiation [11]. However, more than 30% of patients treated through these approaches are likely to exhibit a higher rate of disease recurrence or metastasis [12,13], indicating the difficulty of preventing disease progression. Despite early screening methods and hormone ablation therapies that can effectively reduce the mortality of early-stage PCa, androgen deprivation is not an effective treatment for patients with advanced PCa [11]. Hence, investigating signaling pathways in detail and developing novel targeted therapies to formulate new treatment strategies is of vital importance in enhancing the survival of patients with advanced PCa.

Natural product-derived compounds may represent a promising cancer treatment due to their characteristic of inducing more apoptosis on cancer cells than on normal ones because of their ability to target endoplasmic reticulum (ER) stress-mediated apoptosis in multiple cancer cells [14]. Therefore, they represent an important source of compounds that work against various human diseases including cancer, with approximately half of the pharmaceutical products being developed from traditional herbs [15,16]. More novel natural products with high biological efficiency and low toxicity have been discovered, which hold the potential for new drugs that do not come from compounds derived from total synthesis or combined synthesis, resulting in approximately 40% of drugs developed directly or indirectly from natural products worldwide [17,18]. Indeed, several conventional chemotherapeutic agents such as Taxol, epothilones, and vinca alkaloids are derived from natural products. Natural product-derived compounds have been introduced as a strategy for cancer treatment after being tested against cancer with effective results [14]. Many reports have focused their attention on the molecular signaling pathway modulated by natural products or their derivatives regulating cancer cell apoptosis [14,18,19,20]. Conversely, natural products or their derivatives not only induce cell apoptosis but also lower the resistance to chemotherapeutic agents via modulating apoptosis and metastatic-associated pathways [19,20,21]. Since natural products induce an adaptive pathway and apoptotic pathway, the modulation of apoptosis-associated pathways could represent an effective anti-cancer strategy for treating and preventing cancer.

Tanshinones are purified from the traditional Chinese herb *Salvia miltiorrhiza Bunge* (Danshen) and show various biological activities including angiogenic [22], anti-oxidant [23], and anti-inflammatory effects [24] and are effective against hepatocellular carcinoma [25]. Recent studies have reported that tanshinones can inhibit the growth of PCa cells and gastric cancer through cell death induction [26,27]. In our previous works, a series of novel tanshinone derivatives were designed and synthesized to evaluate their anti-cancer activity (manuscript under submission). A series of tanshinone synthetic derivatives showed inhibitory activity on cancer cell proliferation in vitro by inducing cell apoptosis and arresting the cell cycle. Among these active compounds, 2-((Glycine methyl ester)methyl)-naphtho[1,2-b]furan-4,5-dione (**TC7**, the structure shown in Figure 1A), exhibited the most potent anti-cancer activity with better selectivity and lower toxicity, representing a potential candidate against PCa. Therefore, in this work, the effect of **TC7** was investigated on human PCa cell growth, invasion, and migration, including its molecular mechanisms of action.

## 2. Results 

### 2.1. TC7 Inhibited the Proliferation of PCa Cells and Induced Apoptosis

Several protocols were used to check the effect of **TC7** on the proliferation of PCa cells to determine whether this compound could induce apoptosis in cancer cells (Figure 1). MTT assay results showed that the proliferation of the two PCa cell lines, PC3 and LNCAP, was significantly inhibited by **TC7** treatment in a time- and concentration-dependent manner (Figure 2A). At 48 h, the IC_50_ value of **TC7** on PC3 was 4.11 ± 0.79 μM, and on LNCAP it was 5.62 ± 0.13 μM, both similar to the IC_50_ of doxorubicin used as the positive control (IC_50_ = 3.47 ± 0.43 on PC3, IC_50_ = 4.45 ± 0.81 μM on LNCAP), indicating that **TC7** had a stronger anti-proliferation activity. Further IC_50_ value comparisons at different times between the two cells showed that the inhibitory effect of **TC7** on PC3 proliferation was stronger than that exerted on LNCAP at 12, 24, and 48 h. In addition, the decrease in cell number was concentration-dependent, as shown by the fluorescence microscope images (Figure 2B), since the number of cells decreased significantly as the compound concentration increased. Moreover, some apoptotic bodies were observed in the cells treated with 3, 6, and 12 μmol/L **TC7** at 48 h, suggesting that **TC7** induced apoptosis in PC3 and LNCAP cells. To confirm the inhibitory effect of **TC7** on cell proliferation, a EDU-DNA synthesis assay was performed (Figure 1C). Treatment with 3, 6, and 12 μM **TC7** for 24 h resulted in a dose- and time-dependent proliferation inhibition in both the two cancer cells used. The results showed that the number of cells stained with EDU and Hoechst 33342 decreased with the increase of **TC7** concentration (*p* < 0.01) in both PC3 and LNCAP cells, thus **TC7** suppressed viability and proliferation of human PCa cells by inducing DNA synthesis-dependent apoptosis.

Flow cytometry was used to detect the apoptotic effect of **TC7** on the two PCa cells. Flow cytometry assay results (Figure 1D) showed that apoptosis in PC3 cells treated with 3 μM **TC7** was not significant compared to the control, but 6 and 12 μM **TC7** resulted in a remarkably significant apoptosis (*p* < 0.01) compared with the control group. Apoptosis in LNCAP cells treated with 3 and 6 μM **TC7** was not significant compared to the control, but apoptosis was also remarkably significant (*p* < 0.01) in the cells treated with 12 μM **TC7** compared with that in the control group. The analysis of the ratio of early and late apoptosis showed that apoptosis due to higher concentrations of **TC7** was mainly late apoptosis in both the two cancer cells. These results revealed a certain difference in the apoptotic effect of **TC7** on the two PCa cells used, but the higher concentration of **TC7** induced apoptosis in both. 

To explore the mechanism of **TC7**-induced apoptosis, the expression of key apoptosis-associated proteins was assayed by western blotting in **TC7**-treated PCa cells (Figure 2). The results showed that **TC7** significantly up-regulated p53 (*p* < 0.01), p38 (*p* < 0.01), BAX (*p* < 0.01) and cleaved caspase-8 (*p* < 0.01), while down-regulated ERK1 (*p* < 0.01) and PARP1 (*p* < 0.01) protein expression in PC3 cells compared with the control (DMSO treatment), and significantly decreased caspase-8 expression when treated with **TC7** at high concentration (12 μM) compared to the control (*p* < 0.05), while the increase/decrease of BCL-2 expression was not significant. However, in LNCAP cells, **TC7** showed some differences in the regulation of the expression of the same proteins mentioned above. Indeed, **TC7** significantly increased p53 (*p* < 0.01), caspase-8 (*p* < 0.01), cleaved caspase-8 (*p* < 0.01), and decreased PARP1, ERK1 (*p* < 0.05) protein expression, and significantly decreased BCL-2 expression (*p* < 0.05) when treated with a high concentration of **TC7** (12 μM) compared to the control, but the increase/decrease of p38 and BAX expression was not significant. In addition, **TC7** effect on the phosphorylation of two proteins associated to apoptosis was evaluated in the two cancer cells used. The results showed that ERK1 phosphorylation (p-ERK1/2) was significantly decreased when PC3 cells were treated with a high **TC7** concentration (12 μM) (*p* < 0.01), while p38 phosphorylation (p-p38) was significantly decreased (*p* < 0.01) when PC3 cells were treated with a low **TC7** concentration (3–12 μM) compared to the control. In LNCAP cells treated with **TC7** concentration, the phosphorylation of the two proteins was significantly decreased compared to the control (*p* < 0.01). In summary, **TC7** induced apoptosis of PCa cells through the regulation of key apoptotic proteins such as p53, ERK1, BAX, p38, BCL-2, caspase-8, cleaved caspase-8, and PARP1 as well as influencing the phosphorylation of ERK1 and p38 in the p38/p53/caspase 8 signaling pathway.

### 2.2. TC7 Arrested the Cell Cycle at G2/M Phase in PCa Cells

Flow cytometry was used to detect the effect of **TC7** on the cell cycle of the two types of PCa cells used (Figure 3). The results showed that different concentrations of **TC7** (3, 6, 12 μM) blocked the G1 phase and prolonged the G2 phase (*p* < 0.01), while no significant change was observed in the S phase in PC3 cells compared to the phases in the control cells treated with DMSO. However, only the G2 phase was prolonged in LNCAP cells treated with **TC7** at the same concentrations (*p* < 0.01) compared to the control cells, while no significant difference in the G1 and S phases was observed (Figure 3A). These results demonstrated that the arrest occurred at the G2/M phase in PCa cells incubated for 24 h with **TC7**. To investigate the underlying mechanism of **TC7**-induced cell cycle arrest, the expression of CDC2, cyclin b1, GADD45A, and PLK1 proteins involved in the G2/M cell cycle checkpoint (Figure 3B) were evaluated. Our results showed that **TC7** reduced CDC2 and cyclin b1 protein expression, suggesting that **TC7** acted by suppressing the CDC2/cyclin b1 complex in PC3 cells. However, the expression of CDC2 was increased in LNCAP cells only at the highest **TC7** concentration (12 μM), while cyclin b1 protein expression was decreased by **TC7** in a dose-dependent manner. In addition, PLK1 expression was decreased and GADD45A was increased by **TC7** in a dose-dependent manner in PC3 cells, while PLK1 protein expression was decreased and GADD45A was increased by **TC7** in LNCAP cells, although not in a dose-dependent manner. Since p53 plays an important role in the regulation of the cell cycle, we also evaluated its protein expression. Our results showed that p53 expression was increased by **TC7** in the two cells used (Figure 2B), suggesting that cell cycle arrest by **TC7** was dependent on p53. Therefore, our results indicated that **TC7** arrested the cell cycle at G2/M by regulating the p38/cyclin b1/CDC2 and p53-dependent GADD45A/PLK1 signaling pathway.

### 2.3. TC7 Inhibited Migration and Invasion Ability of PCa Cells

Transwell assay was used to determine the effect of **TC7** on the migration and invasion ability of PCa cells (Figure 4). The experimental results showed that the number of PC3 and LNCAP migrating cells after treatment with **TC7** at different concentrations gradually decreased with the increase of **TC7** concentration (Figure 4B). Indeed, **TC7** significantly inhibited the migration ability of PCa cells compared with the control group (*p* < 0.01). The comparison between PC3 and LNCAP cells revealed that the migration potential of PC3 cells was much higher than that of LNCAP cells (*p* < 0.01). However, all **TC7** doses were able to reduce the migration ability of both the two cell types used, although at the **TC7** dose of 3 μM the difference in migration ability between PC3 and LNCAP was still evident, with PC3 cells migrating more massively that LNCAP. Despite this difference, **TC7** was able to almost completely inhibit the migration of both cell types at the highest dose of 12 μM. This result suggested that **TC7** has the potential to control metastasis of cancer cells. Regarding the invasion ability (Figure 4A), the results are similar, with a similar difference in invasion ability between the two cell types and a similar effect of **TC7** on invasion ability, being able to almost completely inhibit the invasion of both cell types at the highest dose. In order to reduce the effect of apoptosis on the invasion and migration of cancer cells, we investigated the effect of lower concentrations of **TC7** at relatively short treatment times (to ensure that the cells were not induced apoptosis) on the invasion and migration of cancer cells (Figure 4C,D); the results showed that the invasion and migration of the cells was also inhibited (*p* < 0.01) by the lower concentrations of **TC7** at 24 h and the higher concentrations of **TC7** at 12 h. The underlying mechanism used by **TC7** to inhibit migration and invasion of PCa cells was also evaluated by checking the protein expression of MMP-1, MMP-9, and VEGF-1 (Figure 4E). The results showed that **TC7** up-regulated MMP-9 (*p* < 0.01) and down-regulated VEGF-1 (*p* < 0.01) protein expression in both the two types of cells. However, MMP-1 expression was not changed in **TC7**-treated cells. It must be noted that MMP-9 expression was indeed significantly down-regulated (*p* < 0.05) in PC3 cells treated with lower **TC7** concentration (3 μmol/L). Taken together, our results indicate that **TC7** inhibited the migration and invasion ability of PCa cells by regulating the VEGF1/MMP-9 signaling pathway.

### 2.4. **TC7** Inhibited the Proliferation of Human PCa Cells in Vivo

The xenotopic tumor model was obtained by a hypodermic injection of PC3 cells in nude mice to prove anti-PCa activity of **TC7** in vivo (Figure 5). The results showed that the tumor volume in the group treated with **TC7** every two days decreased in size with the increase of the treatment time starting from day 10 compared with the control group (*p* < 0.05 and *p* < 0.01) (Figure 3B), although no significant difference in the tumor volume in the first eight days was observed. After tumor removal, width and length were measured (Figure 5C). The results showed that the tumor weight in the **TC7**-treated mice after 18 days treatment with **TC7** was significantly smaller than that of the control group (*p* < 0.01) (Figure 3D), suggesting a potential anti-tumor activity in vivo. The result of body weight in all nude mice injected with PC3 cells before being sacrificed showed no significant difference between the **TC7**-treated mice and the control group, indicating that the drug was exerting a low toxicity in vivo (Figure 5E). In addition, the main organs, such as heart, liver, spleen, lung, and kidney were collected from the tumor-bearing mice and weighted (Figure 5F). The results showed no significant differences in the weight of all the organs between the treated group and the control group, suggested that **TC7** had no significant effect on the main functional organs in vivo.

## 3. Discussion

In this study, we investigated the anti-tumor effect of the novel tanshinone compound **TC7** in the human PCa cells PC3 and LNCAP, and the underlying molecular mechanisms were also explored by the evaluation of the expression of key proteins and the signaling pathway in which they were involved. Our results in vitro showed that **TC7** inhibited cell proliferation by arresting the cell cycle at G2/M through the regulation of cyclin b1, p53, GADD45A, PLK1, and CDC2/cyclin b1. In addition, **TC7** induced cell apoptosis by regulating apoptosis-associated genes such as p53, ERK1, BAX, p38, BCL-2, caspase-8, cleaved caspase-8, and PARP1, and the phosphorylation level of ERK1 and p38. Furthermore, it decreased DNA synthesis, induced cell apoptosis, and inhibited the migration and invasion ability of PC3 and LNCAP cells by regulating VEGF-1 and MMP-9 protein expression. Our in vivo results clearly demonstrated **TC7**’s ability to reduce tumor volume in tumor-bearing mice. Taken together, our results suggested that the novel tanshinone compound **TC7** could regulate growth, cell cycle, migration, and invasion of the PCa cells through different pathways and might be considered a promising candidate for anti-tumor drug development (Figure 6).

Cell apoptosis can also be induced by the activation of the protein kinase signaling pathway by regulating the expression of mitogen kinase p38 and its phosphorylation (p-p38) that occurs throughout the whole cytosol and close to the cell membrane [28,29,30,31]. p38 has largely been associated with anti-proliferative functions, but several observations indicated that it may also have positive effects on proliferation. Inhibition of the phosphorylation of p38 by inhibitors can prevent TGFβ-induced cell death, implying that the phosphorylation of p38 is crucial in compromising cell apoptosis [32,33,34]. Our results showed that **TC7** significantly inhibited the expression and phosphorylation level of p38 in PC3 cells, while in LNCAP cells it only regulated the phosphorylation level of p38 but not its protein expression, suggesting that the regulation of p38 phosphorylation was the main molecular mechanism of **TC7** inducing cell apoptosis. Furthermore, the sustained ERK1 phosphorylation is suppressed by p38, induced by the growth factors EGF and HGF in Panc-1 cells to control cell proliferation, due to differential cross-talk between the p38 and MEK/ERK pathways [35]. The oncogenic protein ERK1 cascade regulates numerous cellular processes including cell proliferation, differentiation, and survival by relaying extracellular signals and transmitting them throughout the cell [36]. Phosphorylation of ERK1/2 activates a wide array of nuclear substrates, including the direct regulation of downstream proteins PARP1 [37,38]. Functional regulation by phosphorylation of the ERK1 (p-ERK1/2) pathway has been extensively confirmed, and it can also directly regulate the mitochondrial-dependent apoptosis pathway [39]. Our results indicate that **TC7** inhibited ERK1 phosphorylation and protein expression in both the two types of PCa cells used, thus, p38-mediated regulation on the expression and phosphorylation of ERK1 might be associated with **TC7**-induced cell apoptosis, providing new insights in the regulation of the ERK cascade and on the consequent development of new chemotherapeutic agents against PCa with a tanshinone chemical structure. 

As demonstrated by the assessment of the mitochondrial-dependent apoptosis pathway, our results revealed that **TC7** could only significantly up-regulate the expression of BAX in PC3 cells, and only a high concentration of **TC7** could regulate the expression of BCL-2 in LNCAP cells. This result indicated that **TC7** could cause a slightly different expression of key apoptotic proteins in the mitochondrion in both PC3 and LNCAP cells, suggesting that the mitochondria-dependent apoptotic pathway might be involved in **TC7**-induced apoptosis. Caspase-8 and its cleaved form are considered to be the executor caspases of mitochondria-dependent apoptosis, and their activation triggers PARP, which eventually leads to apoptosis [40]. In the intrinsic apoptotic pathway, the immediate translocation of BAX and BCL-2 to the mitochondria subsequently activates caspase-8 and 9 expression both in HeLa and in SW480 cells [38,41]. Our results also confirmed that caspase-8 and cleaved caspase-8 did show a significant increase in **TC7**-treated LNCAP cells, suggesting that caspase-8 and cleaved caspase-8 might be involved in the apoptotic mechanism, although other alternative mechanisms regulating caspase-8 and cleaved caspase-8 were not explored. 

Furthermore, some studies have demonstrated that p38-induced protein expression and phosphorylation of p53 cause its disassociation from MDM2 and consequent evasion of ubiquitin-proteasomal degradation, suggesting that p38 plays a critical role in the modulation of p53 expression in a post-translational manner [42,43,44,45,46,47]. Our results also showed that **TC7** significantly regulated the expression and phosphorylation level of p38 in PC3 cells, while in LNCAP cells only its phosphorylation level was significantly regulated. In addition, **TC7** resulted in a significant increase in the p53 protein level in the two cells. These results indicated that the regulation in the protein expression and phosphorylation of p38 could significantly regulate p53 protein expression. However, a significant difference was found in the regulation of p38 between the two types of cells treated by **TC7**, and the reason for this difference was unclear. In addition, evidence revealed that the inhibition of PARP1, as an important DNA repair factor, may cause the loss of function of the p53 pathway, indicating that PARP1 is regulated by p53 [48]. Our results showed that PARP1 protein expression was significantly decreased by **TC7** in the two PCa cells used. A study showed that the significant regulation in caspase-8 activity and expression temporally coincides with an increase in p53 expression in p53-non-mutated HeLa and SK-HEP-1 cells upon G-Rh2 treatment [38]. Therefore, p53-mediated subsequent downstream caspase-8 activation might contribute to the activation of the downstream DNA repair factor PARP1, leading to tumor cell death. 

Some studies indicated that the inhibition of p38 protein expression can strongly regulate the basal DNA synthesis at the G2 phase and the expression of some cell cycle-associated genes involved in mitosis [35,49,50]. Our results suggested that **TC7** regulated the expression of CDC2 and GADD45A to arrest the cell cycle at the G2 phase. CDC2 as a cyclin protein is involved in the regulation of the transition of the G2 phase into the M phase in the interphase of mitosis [51,52]. The high expression of CDC2, found in many malignant tumors, lead to the abnormal regulation of the cell cycle that is associated with the pathogenesis and progression of cancer [51,53,54,55,56]. Cell cycle progression and arrest are regulated by GADD45 to control various cell fate decisions and functions, such as survival, death, DNA repair, and epigenetic modifications [57,58]. The expression of PLK1 is regulated by p53-dependent transcriptional repression, which plays a major role in driving mitotic events, including centrosome disjunction and separation and is frequently over-expressed in human cancers [59]. PLK1 inhibition is a promising therapeutic strategy and works by arresting cells in mitosis due to monopolar spindles [59]. Our results also showed that **TC7** significantly regulated the expression of PLK1 and p53 in the two PCa cells used, indicating that cell cycle arrest by **TC7** could be related to the regulation of PLK1 by mediating the protein expression of p53. G2/M progression is rigorously controlled by the cyclin b1/CDC2 kinase complex [60,61,62,63]. We also found that the protein expression of the cyclin b1/CDC2 kinase complex was regulated by **TC7** in the two PCa cells used, thereby suggesting that **TC7** arrested the cell cycle at G2/M by regulating the p38/cyclin b1/CDC2 and p53-dependent GADD45A/PLK1 signaling pathways.

The potential molecular mechanism followed by **TC7** to regulate the metastatic ability of PCa cells was also evaluated. Our results indicated that **TC7** regulated VEGF-1 and MMP-9 protein expression in the two PCa cells. A study indicated that inducing up-regulation of VEGF-1 promotes medulloblastoma cell migration and invasion through VEGFR2 signaling, and conversely, inhibition of VEGF-1 expression by specific inhibitors significantly inhibited the invasion and migration of medulloblastoma cells [61,62]. MMP-9 plays vital roles in cancer cell invasion and tumor metastasis, and regulates the expression and release of VEGF-1, which acts as a chemoattractant for invasion in the development of specific inhibitors [63,64]. Furtermore, regulation of p38 phosphorylation may potentially interfere with the function and upstream regulatory genes of VEGF-1 [65], indicating that the molecular mechanism of **TC7** on migration and invasion of PCa cells might be associated with the regulation of the VEGF-1/MMP-9 signaling pathway that mediates by phosphorylation of p38. 

## 4. Materials and Methods

### 4.1. Ethical Issue

All animal experiments were conducted in accordance with the criteria specified in the Guide for the Nursing and Use of Laboratory Animals. The experimental protocols were approved by the Institutional Animal Care Committee at Guizhou medical University (Guizhou, China; approval ID: IACUC 1900361, validity period: 22 March 2018 to 22 March 2020).

### 4.2. Cell Culture

All equipment and biological agents used in this study were the same as previously reported [66,67]. Other reagents were of analytical grade. The human PCa cells PC3 and LNCAP were stored in the Biology laboratory of the Key Laboratory of Chemistry for Natural Products of Guizhou Province and the Chinese Academy of Sciences (Guiyang, China). Cells were cultured in DMEM supplemented with 10% fetal bovine serum (FBS) and 1% penicillin and streptomycin (Sijiqing, Hangzhou, China) and incubated at 37 °C under 5% CO_2_, 95% air, and 95% humidity.

### 4.3. TC7 Compound

The intermediate 2,3-dihydro-2-naphthol[1,2-b]furan-4,5-dione was prepared using 2-hydroxy-1,4-naphthoquinone as starting material via two steps according to previously reported methods [68]. Briefly, treatment of 2-hydroxy-1,4-naphthoquinone with allyl bromide resulted in the formation of the *O*-allyl derivative. The subsequent Claisen rearrangement allowed the generation of allyl lawsone, which was then cyclized to ortho-quinone in the presence of Lewis acid NbCl_5_ at room temperature. 2,3-dihydro-2-naphthol[1,2-b] furan-4,5-dione was reacted with NBS and AIBN to obtain 2-bromomethyl-3-naphtho[1,2-b] furan-4,5-dione, which was then treated with glycine methyl ester hydrochloride overnight at 30 °C and purified by column chromatography to obtain **TC7**. The structure of **TC7** was determined by using high-resolution mass spectra (HRMS) and nuclear magnetic resonance (NMR) spectra. 

### 4.4. Antibodies

All antibodies were purchased from Cell Signaling Technology (Beverly, MA, USA) and used as follows: BAX (no. ab32503, diluted: 1:1000), BCL-2 (no. ab32124, diluted: 1:1000), CASPASE8 (no. ab32397, diluted: 1:1000), PARP1 (no. ab32138, diluted: 1:1000), GADD45A (no. ab203090, diluted: 1:500), p38/MAPK (no. ab170099, diluted:1:1000), phospho-p38MAPK (T180 + Y182) (no. ab195049, diluted: 1:1000), ERK1/2 (no. ab184699, diluted: 1:1000), phospho-ERK 1/2 (T202 + Y204) (no. ab223500, diluted: 1:1000) were purchased from Abcam (Cambridge, USA). Cleaved-caspase 8 (no. 8592, diluted: 1:1000), GAPDH (no. 5174), CDC2 (no. 28439, diluted: 1:1000), CYCLINB1 (no. 4138, diluted: 1:1000), PLK1 (no. 4535, diluted: 1:1000), MMP-1 (no. ab137332, diluted: 1:1000), MMP-9 (no. 13667s, diluted: 1:1000), VEGF-1 (no. 2893s, diluted: 1:1000). Anti-rabbit and anti-mouse LgG (H + L) (Dylight (TM) 800 4×PEG Conjugate) secondary antibodies used in this study were purchased from Cell Signaling Technology and used at a 1:30000 dilution in the experiments. 

### 4.5. MTT Assay

Cell proliferation assay was performed as previously described [67]. Briefly, 6 × 10^3^ cells were cultured in a 96-well plate and incubated until reaching 70% confluence. Next, they were treated with **TC7** at different concentrations (0, 1.25, 2.5, 5, 10, 20 μM/L) with 0.1% DMSO as the control and were then incubated for 12, 24, and 48 h. The experiment was performed in pentaplicate. The same concentrations of doxorubicin were used as the positive control. After treatment, cell morphology was observed by inverted fluorescence microscope equipped with a CCD camera (Nikon, Tokyo, Japan). MTT solution was added to each well, and cells were incubated at 37 °C for 4 h. Then, DMSO was added and plates were incubated at 37 °C for 30 min in a shaking table. The absorbance was read at 490 nm by a Microplate Reader (Thermo Scientific, Vario Skan Flash, NY, USA) and IC_50_ (half maximal inhibitory concentration) values were calculated by SPSS18.0 software (SPASS, Los Angeles, CA, USA).

### 4.6. EDU-DNA Synthesis Assay

The effect of **TC7** on DNA synthesis in the cells was evaluated using 5-ethynyl-2′-deoxyuridine (EDU)-DNA synthesis assay based on a Cell-Light™ EdU Apollo^®^567 in Vitro Imaging Kit (RiboBio, Guangzhou, China) [69]. Briefly, 1 × 10^4^ cells per well were seeded in 96-well plates and incubated for 24 h. Next, **TC7** was added at different concentrations (3, 6, and 12 μM) and cells were incubated for an additional 24 h. Fifty μM EDU were then added to the culture medium and cells were incubated for 2 h, then fixed in 4% paraformaldehyde for 30 min, permeated with 0.5% Trixon-X 100 for 10 min, and stained with 10μM Apollo567 for 30 min. Cells were then counterstained with Hoechst 33342 for 30 min and imaged by high content imaging system (LEICA, DM2000, Munich, Germany).

### 4.7. Invasion and Migration Assay

Invasion and migration ability were evaluated using the same technique [28,67]. The only difference between the two was that in the invasion assay a layer of matrix gel was used to simulate the cell membrane.

Cell invasion ability was determined using transwell chambers (8 μm pores, Corning, Chicago, IL, USA). The upper chamber was coated with 100 μL Matrigel^®^ Matrix 10% (Invitrogen, Carlsbad, CA, USA) and incubated at 37 °C for 4 h, and then 6 × 10^4^ cells were seeded in the upper chamber with 290 μL Opti culture medium. Next, 10 μL **TC7** was added to the medium in the upper chamber at a final concentration of 3, 10, and 12 μM, while 800 μL 1640 medium containing 20% FBS was added to the lower chamber. Cells were incubated at 37 °C for 24 h. Cells were fixed in 75% ethyl alcohol for 10 min, stained with 0.1% crystal violet for 30 min, and rinsed several times with water. The membrane was peeled off using small forceps and allowed to dry. Three random fields were selected to count the cells that invaded the opposite side of the membrane to evaluate the effect of **TC7** on the invasion ability of the cells.

Regarding the migration, transwell chambers were used and 800 μL of 1640 medium containing 20% FBS was added to the lower chamber, while 6 × 10^4^ cells were seeded in the upper chamber with 290 μL Opti culture medium and treated with **TC7** as described in the invasion assay. After 24 h, cells were fixed in 75% ethyl alcohol for 10 min, stained with 0.1% crystal violet for 30 min and rinsed several times with water. The membrane was peeled off using small forceps and allowed to dry. Five random fields were selected to count the cells that migrated on the opposite side of the membrane.

### 4.8. Cell Cycle Assay

The cell cycle assay was performed as previously described [67]. Briefly, cells treated with **TC7** at different concentrations (3, 6, 12 μM) for 24 h were trypsinized and washed with PBS. To avoid cell clumping, cells were fixed by ice-cold ethanol and blocked for 1 h at 4 °C in 5% BSA/PBS. Cells were then re-suspended in PBS containing 1 μg/mL RNase and incubated at 37 °C for 30 min. Propidium iodide (PI) was added to a final concentration of 20 μg/mL, and all cells were moved to round-bottomed 96-well plates and analyzed by flow cytometry using FACS Array (Becton Dickinson, Franklin Lakes, NJ, USA). Histograms of PI signal intensity were generated and the percentage of cells in the G0, G1, S, and G2/M phases were determined.

### 4.9. Cell Apoptosis Assay

Cell apoptosis was detected by Flow Cytometry using Annexin V-fluorescein isothiocyanate (FITC) and propidium iodide (PI) staining kit (BD Pharmingen, San Diego, CA, USA) [67]. In brief, 5 × 10^6^ cells were seeded in 6-well plates, incubated for 12 h, and then treated with **TC7** at different concentrations (3, 6, 12 μM) for 24 h. DMSO 1% was used as the control. Next, cells were digested with trypsin without EDTA, transferred into a centrifuge tube and centrifuged at 1000 rpm for 5 min at room temperature. After washing twice with PBS, cells were re-suspended in 1 × binding buffer, and 100 μL of the cell suspension was transferred to a microcentrifuge tube. PI (Sigma, St. Louis, MO, USA) was then added to a final concentration of 20 μg/mL. Cells were transferred to a round-bottomed 96-well plate and analyzed by flow cytometry (Becton Dickinson, Franklin Lakes, NJ, USA). Apoptotic cells were defined as Annexin-V-positive and apoptosis was expressed as a percentage. 

### 4.10. Western Blot Assay 

**TC7**-treated cells were seeded and treated according to the protocol described in the description of the MTT assay. The treated cells were harvested, washed twice with PBS, and RIPA lysis buffer (60 mL per 10^6^ cells) was added (Solarbio, Beijing, China) and the mixture was incubated at 4 °C for 30 min. The supernatant containing the proteins was obtained by centrifugation at 12,000 rpm for 15 min at 4 °C and protein concentration was measured by BCA Protein Assay Kit (Solaibio, Beijin, China). Sixty μg of proteins from each sample were separated on 10% SDS-PAGE and transferred to a polyvinylidene fluoride (PVDF) membrane (0.22 μm, Merck KGaA, Darmstadt, Germany). The PVDF membrane was sealed in Tris-buffer saline containing 5% non-fat milk and incubated for 1 h at room temperature. Next, the membrane was incubated overnight with the first antibodies at 4 °C. The membrane was washed with TBST and incubated with goat anti-rabbit IgG H&L secondary antibody. GAPDH was used as the loading control. Immunoreactive proteins were tested by an Odyssey Infrared Imaging System. 

### 4.11. Mouse Xenograft Tumor Model and TC7Antitumor Effect

Six week-old male BALB/c-nu nude mice (Beijing Vital River Laboratory Animal Technology Co., Ltd., Beijing, China), weighing 15–18 g, were housed in a controlled environment with a 12h light/dark cycle and with food and water ad libitum. PC3 cells (1 × 10^7^ in 0.1 mL of saline solution) were subcutaneously injected into the right flank of each mouse. After 20 days, when the tumor tissue growth reached approximately 100 mm^3^, mice were randomly divided into two groups (*n* =10) such as the vehicle (DMSO) and the **TC7** treated group. DMSO 0.1% in 25 μL saline solution and **TC7** (60 mg/kg) solution in 25 μL containing 0.1% DMSO were injected into the same right flank every two days for a total of 18 days. Mice weight and tumor size (length and width) were measured every two days to analyze cancer growth rate in vivo. The tumor volume (*V*) in mm^3^ was calculated by the following formula: *V* = 0.5 × a × b^2^ (a: width, b: length) [69].

### 4.12. Statistical Analysis

Statistical analysis was performed using SPSS 13 software package (IBM, Endicott, NY, USA). Results were expressed as mean ± standard deviation (SD) from at least three separate experiments. The Student *t*-test was used to evaluate the significance between the two groups, while multiple-group comparisons were evaluated by one-way ANOVA with post-hoc testing. A value of *p* < 0.05 was considered statistically significant, asterisks show significant difference between groups as * *p* < 0.05, ** *p* < 0.01.

## 5. Conclusions

In conclusion, a new tanshinone analog was investigated by its ability to exert anti-tumor effects on human PCa cells. Our results showed its ability to induce DNA synthesis-dependent apoptosis, cell cycle arrest at the G2 phase, and metastatic capability control by regulating the p38/cyclin b1/CDC2 and p53-dependent GADD45A/PLK1, p38/p53/caspase 8, and p38/VEGF1/MMP-9 signaling pathways. This body of evidence supports the potential use of **TC7** as a promising chemotherapeutic candidate in the treatment of PCa.

## Figures and Tables

**Figure 1 ijms-20-04459-f001:**
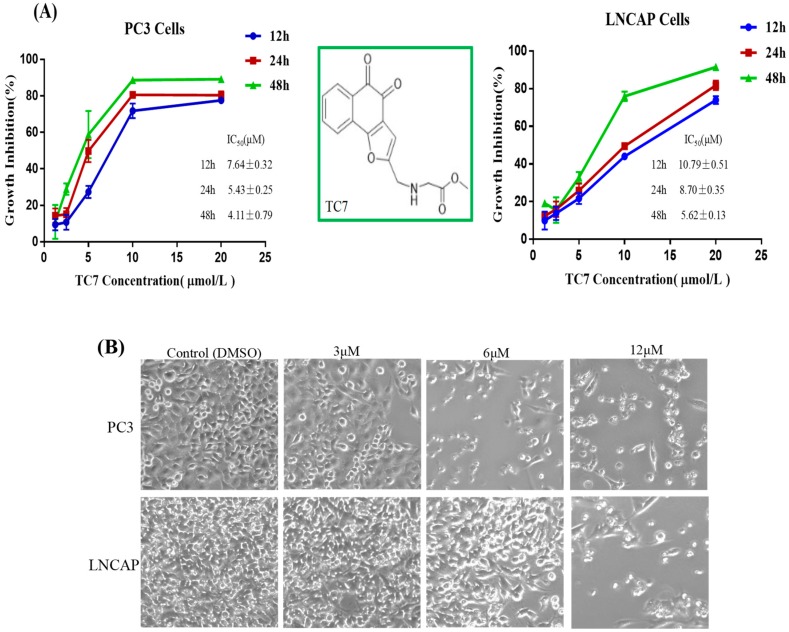
Effects of 2-((Glycine methyl ester)methyl)-naphtho[1,2-b]furan-4,5-dione (**TC7**) on PCa cell growth and apoptosis. (**A**) Growth inhibition induced by **TC7** on PC3 and LNCAP cells by MTT assay. IC_50_ values (μM) of **TC7** were determined according to these curves at different incubation times. (**B**) Cell number and morphological appearance of the two types of cells treated with **TC7** at 3, 6, and 12 μM observed by a fluorescent inverted microscope after 24 h. (**C**) DNA synthesis inhibition by **TC7** on PCa cells by EDU-DNA assay. The zero-hour image was intended to demonstrate that the cells exhibited the higher level of DNA replication before treatment with **TC7**. (**D**) Cell apoptosis induced by **TC7** by flow cytometry. Scale bar = 100 μM in all images. All experiments were performed in triplicate. Results are presented as mean ± SEM. * *p* < 0.05, ** *p* < 0.01 (*n* = 3) compared with the control.

**Figure 2 ijms-20-04459-f002:**
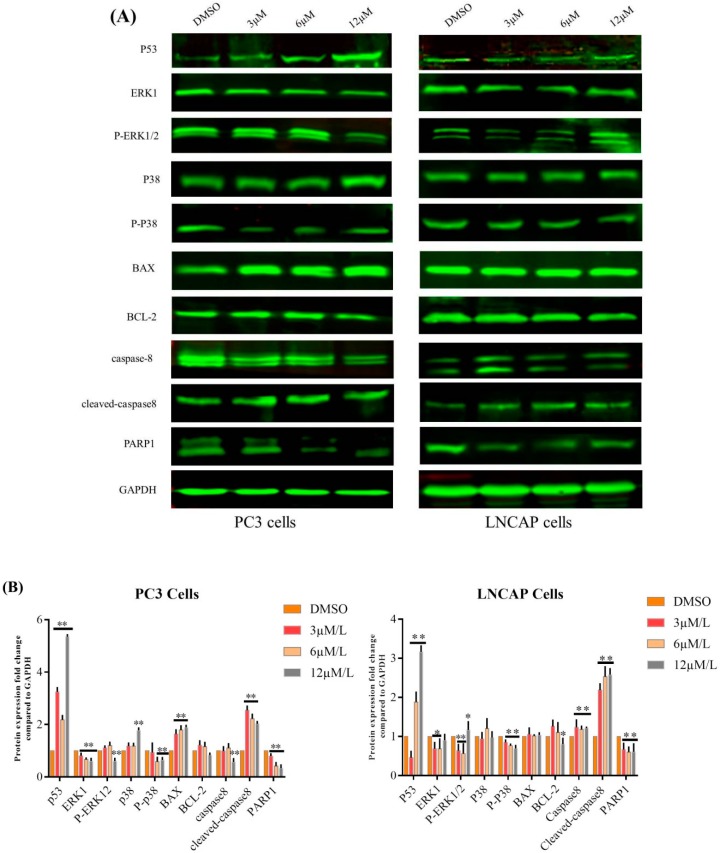
Effects of **TC7** on apoptotic protein expression in PCa and LNCAP cells. Cells were incubated with **TC7** at different concentrations (3, 6, 12 μM) for 24 h, then total proteins were extracted. (**A**) The expression of apoptotic proteins was detected using western blot in the cells treated with **TC7**. (**B**) Quantitative analysis of the protein expression by Image J software. GAPDH was used as a loading control. Experiments were performed in triplicate. Results are presented as mean ± SEM. * *p* < 0.05 (*n* = 3) compared with the control, ** *p* < 0.01 (*n* = 3) compared with the control.

**Figure 3 ijms-20-04459-f003:**
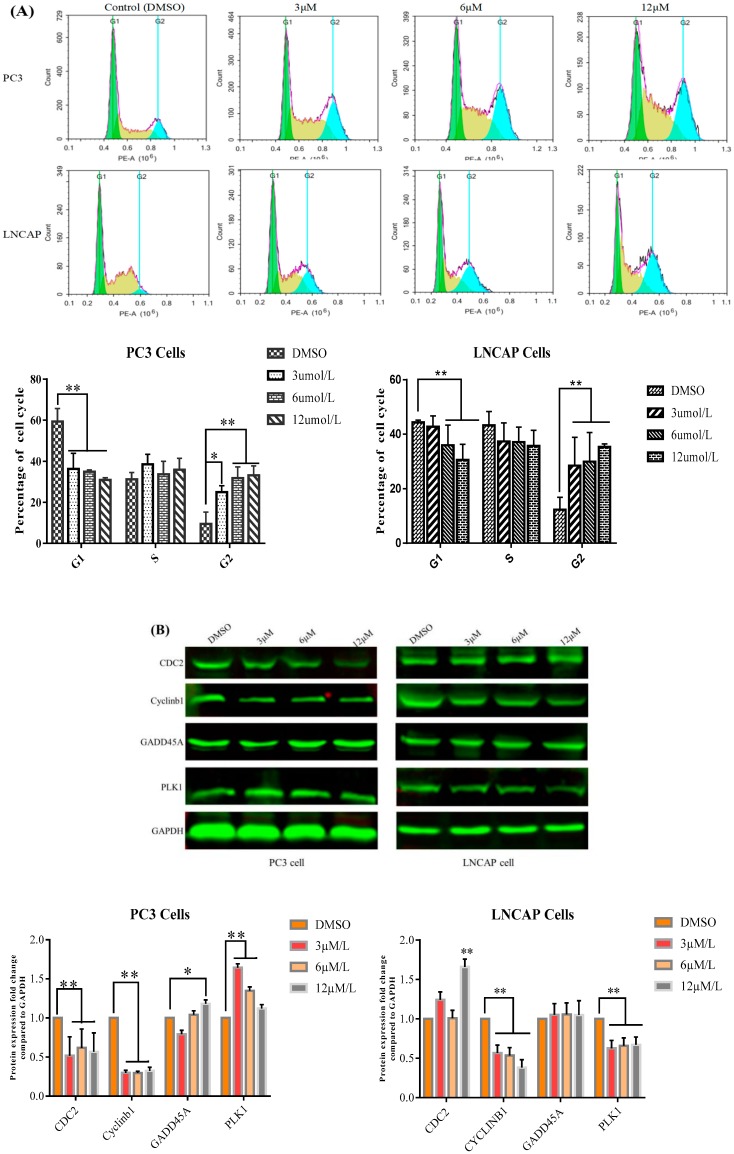
Effect of **TC7** on cell cycle of PCa and LNCAP cells. Cells were treated with **TC7** at different concentrations (3, 6, and 12 μM) for 24 h. (**A**) Effect of **TC7** on the cell cycle of PCa and LNCAP cells by flow cytometry. Cells were fixed and stained with propidium iodide (PI) to analyze the DNA content by flow cytometry. The bar graph shows PC3 and LNCAP cells percentage in different phases of the cell cycle. (**B**) **TC7** effect on the expression of proteins involved in the cell cycle of PCa and LNCAP cells by western blot and quantitatively analyzed by Image J software. Results are presented as mean ± SD from three independent experiments. * *p* < 0.05 (*n* = 3); ** *p* < 0.01 (*n* = 3), compared with the control group.

**Figure 4 ijms-20-04459-f004:**
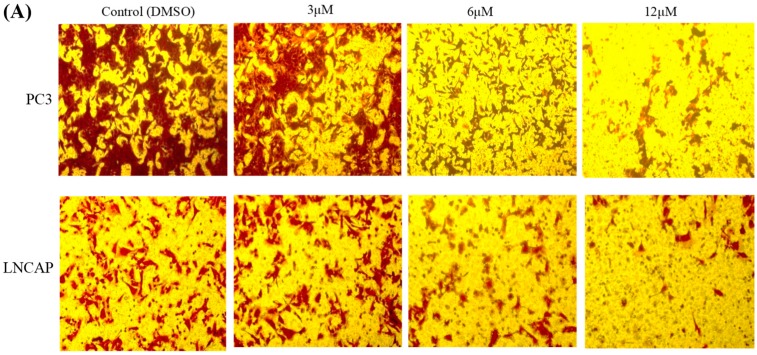
Effect of **TC7** on migration and invasion ability of PCa cells. Migration (**B**) and invasion (**A**) of PC3 and LNCAP cells was evaluated by Transwell assay. Cells were treated with **TC7** at different concentrations (3, 6, and 12 μM) for 24 h. The number of migrated and invading cells were quantified and compared with the DMSO group. Scale bar = 100 μm in all images. (**C**,**D**) Effect of lower concentrations of **TC7** at relatively short treatment times on the invasion and migration of cancer cells. (**E**) Effect of **TC7** on the expression of proteins involved in migration and invasion by western blot and quantified by Image J software. Results are presented as mean ± SD from three independent experiments. * *p* < 0.05 (*n* = 3); ** *p* < 0.01 (*n* = 3), compared with the control group; *## p* < 0.01 (*n* = 3), compared between two cancer cell lines.

**Figure 5 ijms-20-04459-f005:**
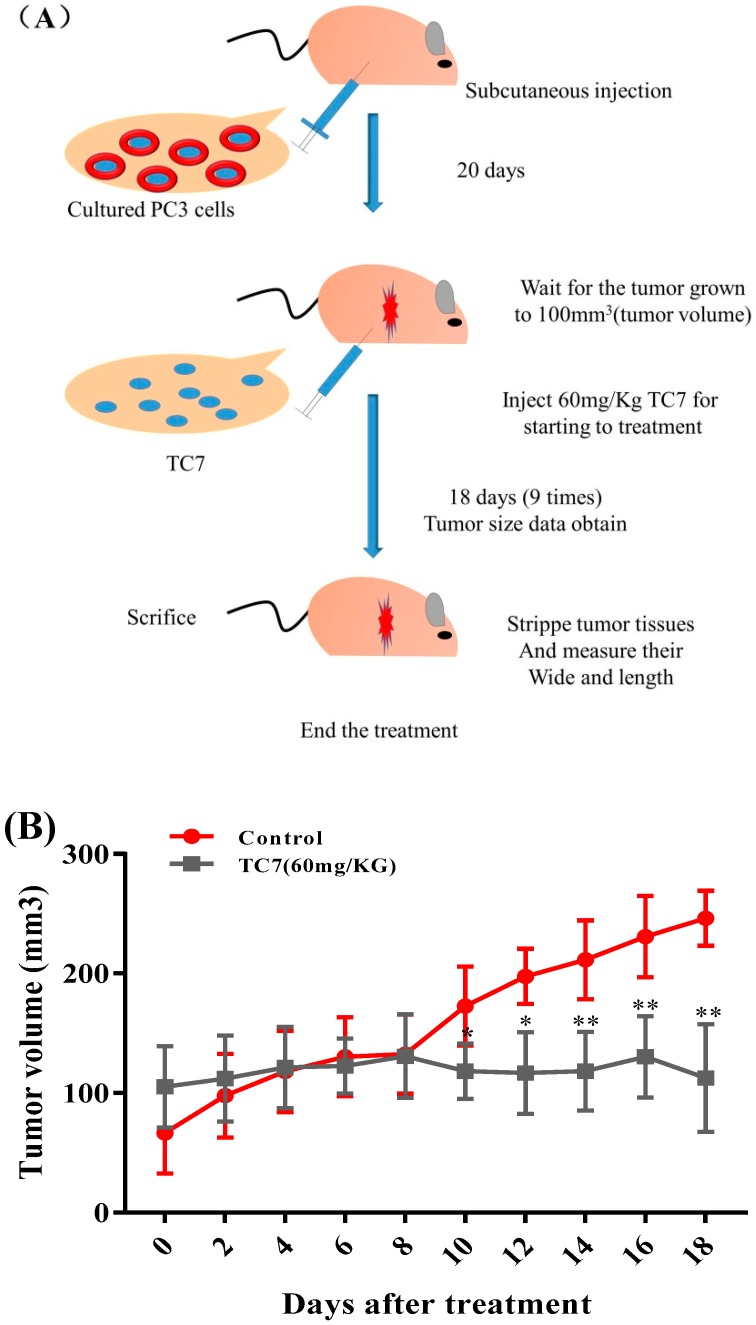
Effect of **TC7** on tumor growth in vivo. (**A**) The xenotopic tumor model in nude mice injected with PC3 cells were divided into two groups; one was intraperitoneally treated with **TC7** (60 mg/kg), and the control group was treated with 0.1% DMSO every two days, nine times in total, as indicated in (**A**). (**B**) The tumor size in the **TC7**-treated and the control group was measured at three-day intervals as soon as **TC7** was injected. (**C**) Tumors from the **TC7**-treated and the control group after 18 days treatment with **TC7**. (**D**) Tumor size in the **TC7**-treated and the control group after 18 days treatment with **TC7**. (**E**) Mice body weight was monitored at two-day intervals as soon as the **TC7** was injected. **(F)** Weight change in the main organs in the mice of the **TC7**-treated and the control group after 18 days treatment with **TC7**. Results are presented as mean ± SD from three independent experiments. * *p* < 0.05 (*n* = 3); ** *p* < 0.01 (*n* = 3), compared with the control group.

**Figure 6 ijms-20-04459-f006:**
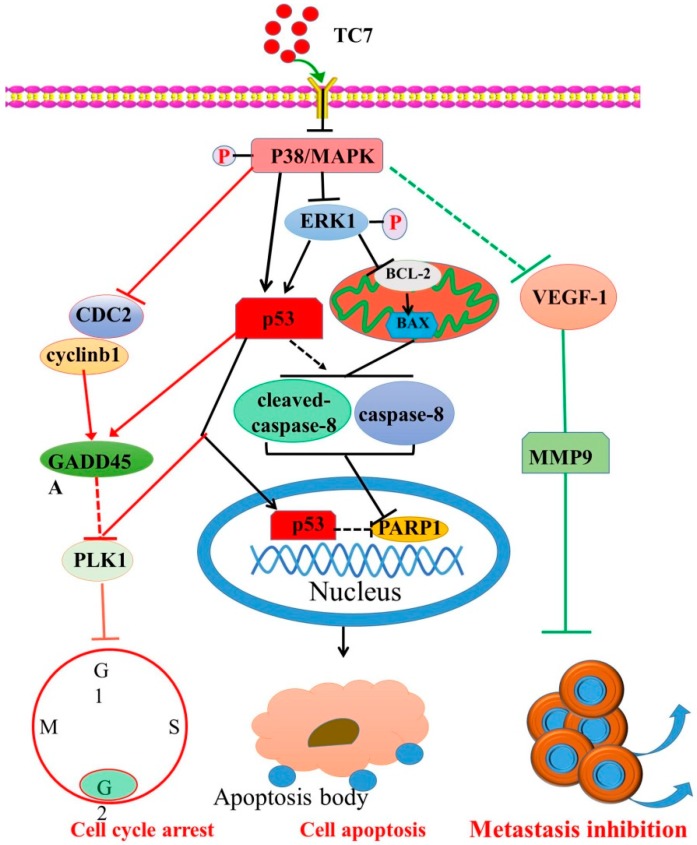
The proposed cell signaling pathways used by **TC7** to induce PCa cell apoptosis, cell cycle arrest, and metastasis inhibition through the regulation of p38/cyclin b1/CDC2, and p53-dependent GADD45A/PLK1, p38/P53/caspase 8, and p38/VEGF-1/MMP-9 signaling pathway.

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
