# Peer review of "A Novel Tanshinone Analog Exerts Anti-Cancer Effects in Prostate Cancer by Inducing Cell Apoptosis, Arresting Cell Cycle at G2 Phase and Blocking Metastatic Ability"

_ijms, 2019, doi:10.3390/ijms20184459_

Round 1

Reviewer 1 Report

The manuscript is clear and well written and the authors demonstrate that growing concentrations of TC7 inhibit cell proliferation, migration and invasion in vitro and reduce the tumor size in vivo. They use two prostate cancer cell lines completely different each other: LNCaP and PC3 and observe two different way to reacts to the TC7 treatment. Is this linked to the differences between these two cell lines? They have to note that the LNCaP cells express the androgen receptor (that play a pivotal role in proliferation, migration and invasion) and the PC3 cells no. Again they use female mice. Is this the right choice? It is possible that in male mice they could observe a different behaviour?

Do the authors speculate a possible receptor used by TC7?

The authors indicate in Fig. 6 the possible pathway used by TC7 in prostate cancer cells but, through the previous experiments they only describe the quantitative variation of different proteins after TC7 treatments and don't use inhibitors or other substances to demonstrate this pathway.

The same is true when they discuss about the difference observed between LNCaP and PC3 cells in p38 phosphorylation level or p38 expression level or about a possible link between p38 and p53. To my humble opinion, they have to demonstrate what they assert. 

In fig 1C, they use the EDU assay that measure the new synthesized DNA but, into the histogram, they indicate the percentage of apoptosis. This is wrong because the EDU assay indicate the number of newly synthesized cells that incorporate EDU into the new DNA and not the cell death.

Into the same figure, when I compare the contrast phase images and the EDU images, I can see that the contrast phase images for LNCa are more similar to the PC3 EDU images and vice-versa. May I see the 0 time of the contrast phase images for each treatment?

Why the authors don't measure the activated and the pre-MMPs proteins but only the MMPs levels? Why they choose the MMP1 and 9 and not the MMP2?

Author Response

1)They use two prostate cancer cell lines completely different each other: LNCaP and PC3 and observe two different way to reacts to the TC7 treatment. Is this linked to the differences between these two cell lines?

Response:As our study involves the regulation of invasion and metastasis of tumor cells by small molecules, we selected a prostate cancer cell with high metastasis ability (PC3) and low metastasis ability (LNCAP) for comparative study according to literature reports (Aalinkeel R, Nair M P N, Sufrin G, et al. Gene expression of angiogenic factors correlates with metastatic potential of prostate cancer cells[J]. Cancer research, 2004, 64(15):5311-5321.), and the results showed that the migration and invasion ability of PC3 cells in the DMSO group was higher than that of LNCAP, and the metastatic ability of two cells was significantly inhibited using the TC7treatment, which indicated that  TC7 has a certain regulatory effect on the invasion and migration of prostate cancer cells with higher and lower metastasis ability. we speculated the molecular mechanism of its regulation may have certain difference, the molecular experiments results in the study also suggested that there was some difference for regulating the expression of metastasis-associated genes in the two prostate cancer cells.

2)They have to note that the LNCaP cells express the androgen receptor (that play a pivotal role in proliferation, migration and invasion) and the PC3 cells no. Again they use female mice. Is this the right choice?

Response:We are very sorry for our negligence in the writing process. In order to avoid the effects of estrogen in female nude mice on prostate cancer cells and to minimize the effects of hormones on the cancer cells, we used immunodeficient male BALB/c nude mice in the experiment.

3) The same is true when they discuss about the difference observed between LNCaP and PC3 cells in p38 phosphorylation level or p38 expression level or about a possible link between p38 and p53. To my humble opinion, they have to demonstrate what they assert. 

Response:Thanks for your suggestion. This article is our first study of the molecular mechanism of this compound, and we speculated that the compound exhibited a potential to regulate p38 pathway according to our experiment results. Our group are studying the regulated mechanism of compounds on the p38 signaling pathway.

4)Do the authors speculate a possible receptor used by TC7?

Response: In this paper, we mainly focused on the evaluation of the anticancer activity of TC7 and preliminarily discussed its molecular mechanisms, while the receptor of TC7 and its related molecular mechanism were not confirmed or speculated. Your comments give us some inspiration. We focus on designing a special project to solve the receptor and its molecular mechanism of the compound, and our research group is now carrying out this research.

5)In fig 1C, they use the EDU assay that measure the new synthesized DNA but, into the histogram, they indicate the percentage of apoptosis. This is wrong because the EDU assay indicate the number of newly synthesized cells that incorporate EDU into the new DNA and not the cell death.

Response: Thanks for your advice. We have already checked the literature to label the pictures correctly (Lu S , Zhixian G , Jihong S , et al. MiR-133a acts as an anti-oncogene in Hepatocellular carcinoma by inhibiting FOSL2 through TGF-β/Smad3 signaling pathway[J]. Biomedicine & Pharmacotherapy, 2018, 107:168-176). In the histogram, we have changed the percentage of apoptosis to EDU%.

6)Into the same figure, when I compare the contrast phase images and the EDU images, I can see that the contrast phase images for LNCa are more similar to the PC3 EDU images and vice-versa. May I see the 0 time of the contrast phase images for each treatment?

Response:Of course. Thank you very much for your questions. When we tested the effect of TC7 on DNA replication in prostate cancer cells, we first needed to inoculate the counted cells into the plate until all cells were seeded to the plate before adding TC7. In fact, when we observed immediately (0h) the extend of DNA replication in cells after adding TC7, it was actually revealed the state of DNA replication when the cells were completely attached to the plate. We could see that the number of cells labeled by EDU in two cell lines at 0h was less than that of DMSO-treated group for 24h, but was more than that of TC7-treated groups, which means that TC7 could affect the DNA replication of cancer cells with time increasing.

7)Why the authors don't measure the activated and the pre-MMPs proteins but only the MMPs levels? Why they choose the MMP1 and 9 and not the MMP2?

Response:In the experimental process of discussing the regulatory mechanism of TC7 on invasion and migration ability of prostate cancer cells, we consulted the relevant literatures, and found that the expression of MMP-1 and MMP-9 was strongly correlated with the invasion and migration of prostate cancer cells in the literatures (Li N , Dhar S S , Chen T Y , et al. JARID1D Is a Suppressor and Prognostic Marker of Prostate Cancer Invasion and Metastasis.[J]. Cancer Research, 2016, 76(4):831.),( Miftakhova R , Hedblom A , Semenas J , et al. Cyclin A1 and P450 Aromatase Promote Metastatic Homing and Growth of Stem-like Prostate Cancer Cells in the Bone Marrow[J]. Cancer Research, 2016, 76(8).), so we chose to study the regulation of TC7 on the expression of MMP-1 and MMP-9 to discuss the molecular mechanism. Due to the limited literature reading, the regulation of mmp-2 expression and activation of the pre-MMPs by TC7 was indeed not considered in our study. We are also very grateful to the reviewers for the comments, which provide a good idea for elucidating the anticancer molecular mechanism of this compound, and we will carry out in-depth research in accordance with the comments.

Reviewer 2 Report

Congratulations on some excellent scientific experiments. Could you please add the information on the LD50 for this new agent? In order to take this to a phase 1 trial one needs to know the dose that could be used in humans. Did you find any drug induced toxicity with the mouse models? Could you please elaborate on the findings from the harvested organs? Apart from the weight we need to know whether any significant toxicities were identified. There is some tumour shrinkage, but more work is needed on the drug's safety profile.

Author Response

(1)Could you please add the information on the LD50 for this new agent? In order to take this to a phase 1 trial one needs to know the dose that could be used in humans.

Response:In this study, we mainly discussed the anti-prostate cancer activity and molecular mechanism of TC7, and its in vivo safety and related dose effect (LD50) were not involved in this study. As the druggability evaluation study of TC7, we will conduct a systematic step by step in-depth study. Thank you very much for giving the opinion. We will complete the in vivo dose-effect, safety and stability study of TC7 as soon as possible, and we will report the druggability of TC7.

(2) Did you find any drug induced toxicity with the mouse models? 

Response: According to our results, we did not find any drug induced toxicity with the mouse models.

(3)Could you please elaborate on the findings from the harvested organs? 

Response: In evaluating the effects of TC7on the growth of prostate cancer cells in mice by local vaccination, we compared the weight change of the heart, liver, spleen, lung and kidney in both control and treated groups, it was found that the weight of the several main organs were no significant differences, the preliminary showed that local injections of TC7 in mice had no obvious effect on the weight of the major organs, but whether affecting the function of these organs is unsure, our team is studying until now.

Round 2

Reviewer 1 Report

Dears Authors,

I have other doubts about your manuscript. 

In fig. 1C you show that, in both cell lines, the EDU incorporation is reduced after 24h of TC7 treatment (all concentrations). Now, considering that EdU is inncorporated during the S phase, how do you compare these results with the S phase trade obtained after FACS analisys and showed in Fig. 3A? Again, if the cell cycle inhibition is in G2 phase, why the EdU incorporation is reduced?

The authors assert that TC7 treatments are responsible for inhibition of migration and invasion. To this purpose, how do they explain the increase of MMP9 level? Did they also measure the MMP9 levels in conditioned medium?

They measure the number of migrated and invading cells after 24h but, during this time they register an high number of apoptotic cells (late apoptosis). Is possible that migration and invasion data are influenced by apoptosis values?

Author Response

1.In fig. 1C you show that, in both cell lines, the EDU incorporation is reduced after 24h of TC7 treatment (all concentrations). Now, considering that EdU is inncorporated during the S phase, how do you compare these results with the S phase trade obtained after FACS analisys and showed in Fig. 3A? Again, if the cell cycle inhibition is in G2 phase, why the EdU incorporation is reduced?

Response:We showed that DNA replication was significantly inhibited in TC7-treated cells by EdU assay, while flow cytometry assay showed that TC7 significantly blocked the G2 phase of the cells. We are known that DNA replication is mainly concentrated in S phase, while G2 phase is mainly the synthesis of proteins related to enzymes and spindle formation in cell division phase. Why these two results are inconsistent, we speculate that TC7 may play a more significant role in regulating the synthesis of proteins related to enzymes and spindle formation during cell division. However, TC7 also has a regulatory effect on DNA replication in this process, but its effect may not be as significant as its effect on S2 phase. In addition, higher concentration of TC7 treatment can significantly induce apoptosis, which can also significantly affect cell cycle changes, which may be the main reason why flow cytometry analysis showed that TC7 significantly blocks cells in G2 phase, we are planning to research this subject more deeply in our present study.

2.The authors assert that TC7 treatments are responsible for inhibition of migration and invasion. To this purpose, how do they explain the increase of MMP9 level? Did they also measure the MMP9 levels in conditioned medium?

Response:In fact, MMP-9 expression was significantly down-regulated in PC3 cells, with high metastatic potential, treated with lower TC7 concentration (3 μM), but the expression was increased with TC7 concentration. However, the expression of MMP-9 was significantly up-regulated in LNCAP cells with low metastasis potential. We speculated that high concentration of TC7 could significantly induce apoptosis and up-regulate the expression of mmp-9. Therefore, we redesigned the supplementary detection for determining the invasion and migration ability of cells under lower concentrations of TC7 and shortened treatment time to ensure that cells were not significantly induced apoptosis, and the results showed that TC7 did significantly inhibit the invasion and migration ability of cells. We also looked at the literature (Amigo-Jiménez I, Bailón E, Aguilera-Montilla N, García-Marco JA, García-Pardo A. Gene expression profile induced by arsenic trioxide in chronic lymphocytic leukemia cells reveals a central role for heme oxygenase-1 in apoptosis and regulation of matrix metalloproteinase-9. Oncotarget. 2016; 7(50):83359–83377. doi:10.18632/oncotarget.13091) that showed the apoptosis cells can up-regulated the expression of MMP-9, which indicated that the significant up-regulation of MMP-9 in prostate cancer cells treated by TC7 in our study may be due to TC7-induced apoptosis.

3.They measure the number of migrated and invading cells after 24h but, during this time they register an high number of apoptotic cells (late apoptosis). Is possible that migration and invasion data are influenced by apoptosis values?

Response: Thank you very much for your advice. It is true that apoptosis can significantly affect the invasion and migration ability of cells. We carefully considered and discussed how to avoid the influence of apoptosis on cell invasion and migration. We redesigned the supplementary detection for determining the invasion and migration ability of cells under lower concentrations of TC7 and shortened treatment time to ensure that cells were not significantly induced apoptosis, and the results showed that TC7 did significantly inhibit the invasion and migration ability of cells. We added the supplemented experimental results in the revised manuscript.

Round 3

Reviewer 1 Report

The authors answered completely to my objection.